# The LIFR Inhibitor EC359 Effectively Targets Type II Endometrial Cancer by Blocking LIF/LIFR Oncogenic Signaling

**DOI:** 10.3390/ijms242417426

**Published:** 2023-12-13

**Authors:** Nicole Spencer, Alondra Lee Rodriguez Sanchez, Rahul Gopalam, Panneerdoss Subbarayalu, Daisy M. Medina, Xue Yang, Paulina Ramirez, Lois Randolph, Emily Jean Aller, Bindu Santhamma, Manjeet K. Rao, Rajeshwar Rao Tekmal, Hareesh B. Nair, Edward R. Kost, Ratna K. Vadlamudi, Suryavathi Viswanadhapalli

**Affiliations:** 1Division of Reproductive Research, Department of Obstetrics and Gynecology, University of Texas Health San Antonio, San Antonio, TX 78229, USA; spencern@uthscsa.edu (N.S.); rodriguezsaa@uthscsa.edu (A.L.R.S.); gopalam@uthscsa.edu (R.G.); yangx1@uthscsa.edu (X.Y.); ramirezp6@uthscsa.edu (P.R.); randolphl@livemail.uthscsa.edu (L.R.); aller@livemail.uthscsa.edu (E.J.A.); tekmal@uthscsa.edu (R.R.T.); kost@uthscsa.edu (E.R.K.); vadlamudi@uthscsa.edu (R.K.V.); 2Department of Cell Systems & Anatomy, Greehey Children’s Cancer Research Institute, University of Texas Health San Antonio, San Antonio, TX 78229, USA; subbarayalu@uthscsa.edu (P.S.); medinad2@livemail.uthscsa.edu (D.M.M.); raom@uthscsa.edu (M.K.R.); 3Evestra, Inc., San Antonio, TX 78245, USA; bsanthamma@evestra.com (B.S.); hnair@evestra.com (H.B.N.); 4Mays Cancer Center, University of Texas Health San Antonio, San Antonio, TX 78229, USA; 5Audie L. Murphy Division, South Texas Veterans Health Care System, San Antonio, TX 78229, USA

**Keywords:** Type II endometrial cancer, EC359, LIFR, LIF, PDX, novel therapy

## Abstract

Endometrial cancer (ECa) is the most common female gynecologic cancer. When comparing the two histological subtypes of endometrial cancer, Type II tumors are biologically more aggressive and have a worse prognosis than Type I tumors. Current treatments for Type II tumors are ineffective, and new targeted therapies are urgently needed. LIFR and its ligand, LIF, have been shown to play a critical role in the progression of multiple solid cancers and therapy resistance. The role of LIF/LIFR in the progression of Type II ECa, on the other hand, is unknown. We investigated the role of LIF/LIFR signaling in Type II ECa and tested the efficacy of EC359, a novel small-molecule LIFR inhibitor, against Type II ECa. The analysis of tumor databases has uncovered a correlation between diminished survival rates and increased expression of leukemia inhibitory factor (LIF), suggesting a potential connection between altered LIF expression and unfavorable overall survival in Type II ECa. The results obtained from cell viability and colony formation assays demonstrated a significant decrease in the growth of Type II ECa LIFR knockdown cells in comparison to vector control cells. Furthermore, in both primary and established Type II ECa cells, pharmacological inhibition of the LIF/LIFR axis with EC359 markedly decreased cell viability, long-term cell survival, and invasion, and promoted apoptosis. Additionally, EC359 treatment reduced the activation of pathways driven by LIF/LIFR, such as AKT, mTOR, and STAT3. Tumor progression was markedly inhibited by EC359 treatment in two different patient-derived xenograft models in vivo and patient-derived organoids ex vivo. Collectively, these results suggest LIFR inhibitor EC359 as a possible new small-molecule therapeutics for the management of Type II ECa.

## 1. Introduction

Endometrial cancer (ECa) is the most common gynecologic malignancy in developed countries [1,2]. There are two main histologic subtypes of ECa, including endometrioid endometrial carcinoma (ECa), or Type I ECa, and non-endometrioid carcinomas, or Type II ECa. About 80% of ECa are endometrioid ECa (EEC) (Type I), with the remaining 20% consisting of mixed ECa, uterine carcinosarcoma (UCS) (Type II), serous ECa (SEC), and clear-cell ECa (CEC) [3]. High-grade ECa includes the histological subtypes of papillary serous carcinoma, clear-cell carcinoma, carcinosarcoma, and certain FIGO Grade 3 endometrioid adenocarcinoma. When compared to individuals with low grade ECa, patients with high-grade ECa are more likely to be diagnosed at a later stage and are more likely to recur [4]. Patients with Type I ECa typically have a favorable prognosis; however, patients with advanced stage, high-grade ECa, and Type II ECa have poorer outcomes [5].

Although progestins have shown promising results in terms of treating patients undergoing uterine-sparing treatment, the addition of adjuvant hormone therapy provides only moderate benefits to Type I ECa patients, with a recurrence rate of approximately 50% [6]. Conversely, patients with Type II ECa experience less favorable outcomes with hormone therapy [5]. In clinical trials, the use of targeted agents such as bevacizumab, everolimus, and metformin did not result in long-term remission in patients with recurrent or advanced ECa [5,7,8]. Furthermore, serous ECa (SEC) accounts for 10% of all ECa and is the most biologically aggressive variant of Type II ECa. Treatment options for advanced Type II ECa are limited and the identification of new targets for treating Type II ECa represents an unmet need [9].

Patients with high-grade ECa have been shown to have elevated serum levels of inflammatory cytokines, including IL6 [10]. Leukemia inhibitory factor (LIF) is a member of the IL6 family of cytokines and acts on the LIF receptor (LIFR) and glycoprotein 130 (gp130) complex [11]. This complex stimulates a signaling cascade and the activation of STAT3, MAPK, AKT, and mTOR, leading to tumor proliferation, metastasis, and therapy resistance in many solid tumor types [12]. Recent studies suggested the oncogenic role of LIF/LIFR signaling in the progression of Type I ECa [13]. Additionally, the inhibition of LIFR signaling blocks progression of obesity-driven Type I ECa [14]. However, the role of LIF/LIFR signaling in the progression of Type II ECa remains unknown.

EC359 is a first-in-class small molecule that acts as an inhibitor on the LIFR [15]. The specificity of EC359 towards LIFR and its functional effects were thoroughly examined using a series of rigorous experiments including knockout cells, biophysical assays, microscale thermophoresis (MST), computational modeling, and pull-down assays. EC359 treatment demonstrated pharmacological benefits, including in oral bioavailability and in vivo stability [13,15].

In this study, we examined the significance of LIF/LIFR signaling in the progression of Type II ECa and evaluated the utility of EC359 for treating Type II ECa. We used rigorous research design, including the use of multiple primary as well as established Type II ECa cells and LIFR knockdown models. The therapeutic efficacy of EC359 was tested in clinically relevant models of Type II ECa, including patient-derived organoids (PDO) and xenografts (PDX). Using primary and established cell lines, we demonstrated that EC359 reduced cell viability and colony formation, and also promoted apoptosis of Type II ECa cells. Mechanistic studies confirmed the activation of LIF/LIFR signaling in Type II ECa cells and attenuation of this signaling was observed with LIFR inhibitor EC359 treatment. Further, using PDOs and PDX models, we showed that treatment with LIFR inhibitor EC359 reduced the in vitro and in vivo progression of Type II ECa. The results of our study suggest that the LIFR inhibitor EC359 has potential as a targeted therapeutic for the treatment of Type II ECa.

## 2. Results

### 2.1. Type II ECa Patients Express Higher Levels of LIF

Analysis using cBioPortal for cancer genomics survival analysis (http://cbioportal.org/ (accessed on 8 October 2023)) indicates that altered LIF expression is associated with poor overall survival in ECa patients (Figure 1A). Among cases with altered LIF expression, 30 out of 40 identified cases belong to patients with Type II ECa (Figure 1B). These data support the possible association of altered LIF expression with poor overall survival in Type II ECa. We also examined LIF status in endometrial carcinosarcoma using the publicly available OncoDB platform, which enables the comparison of gene expression between tumor and normal tissues using validated databases [16]. The results showed that LIF is highly expressed in endometrial carcinosarcoma compared to normal tissues (Figure 1C). Collectively, these results suggest that LIF expression is increased in Type II ECa and is correlated with poor overall survival.

### 2.2. Functional LIFR Is Needed for Optimal Growth of Type II ECa Cells

To provide genetic evidence that intrinsic LIF/LIFR signaling in Type II ECa cells was beneficial to ECa progression, we generated ECa-47 LIFR knockdown (KD) cells using two distinct siRNAs. Western blotting analyses confirmed the KD of LIFR (Figure 1D). Cell viability and colony formation assays showed the reduced growth of Type II ECa LIFR-KD cells compared to vector controls (Figure 1E–G).

### 2.3. EC359 Reduced the Cell Viability, Survival, and Invasion of Patient-Derived and Established Type II ECa Cells

Recently, we developed a small-molecule inhibitor, EC359, to specifically target the LIFR and effectively block LIF/LIFR signaling [15]. We examined the effect of EC359 on the cell viability of patient-derived and established Type II ECa cells. Treatment with EC359 resulted in a significant dose-dependent reduction in the cell viability of ECa cells (IC_50_ ~1–10 nM) (Figure 2A). We next examined the effect of EC359 on the survival of ECa cells. In clonogenic survival assays, EC359 significantly reduced the colony-forming ability of ECa cells (Figure 2B,C). Given the important role of the LIF axis in the invasiveness of cancer cells, we examined the effect of EC359 in terms of reducing the invasion of Type II ECa cells. Matrigel invasion assays demonstrated that EC359 significantly reduced the invasive potential of ECa cells (Figure 2D,E). Collectively, these results suggest that EC359 exhibits significant inhibitory activity on the cell viability, cell survival and invasion of ECa cells.

### 2.4. EC359 Induced Apoptosis and Inhibited the Growth of Patient-Derived Organoids (PDOs)

We next examined whether EC359 promotes apoptosis in ECa cells using the Annexin V staining assay. EC359 treatment significantly increased Annexin V-positive cells (Figure 3A–F) in multiple patient-derived Type II ECa cells. Collectively, these results suggest that EC359 significantly promotes the apoptosis of Type II ECa cells. We next tested the effect of EC359 on the growth of organoids established from primary Type II ECa tissues. Organoid viability analysis of patient-derived organoids indicated that EC359 treatment significantly reduced their viability compared to vehicle treatment in a dose-dependent manner (Figure 3G,H). Collectively, these results suggest that EC359 exhibits significant enhanced activity upon apoptosis and inhibits the growth of PDOs.

### 2.5. EC359 Reduced LIFR Downstream Signaling in Type II ECa Cells

To confirm the inhibitory effect of EC359 on LIF/LIFR-mediated STAT3 activation, Type II ECa cells stably expressing the STAT3-Luc reporter were treated using vehicle or EC359. As expected, EC359 treatment significantly inhibited STAT3 reporter activity (Figure 4A) in both ECa-47 and ECa-81 cells. In RT-qPCR assays using ECa-78 cells, EC359 treatment significantly reduced the expression of several known LIFR target genes (Figure 4B). To further confirm the effect of EC359 on LIF/LIFR downstream signaling pathways, Type II ECa cells were treated with vehicle or EC359 and the status of LIFR downstream signaling was examined using Western blotting. EC359 treatment substantially reduced the activation of STAT3 and its downstream targets including pAkt, pMAPK, and pmTOR in ECa-47 and KLE cells (Figure 4C). Overall, these findings suggest that EC359 is effective in blocking LIF/LIFR constitutive signaling in Type II ECa cells.

### 2.6. EC359 Reduced Patient-Derived Xenograft (PDX) Tumor Growth In Vivo

To test the efficacy of EC359 on in vivo tumor progression of Type II ECa, we used two different PDX tumors (ECa-15 and 81) in the mice. When the tumor reached ~200 mm^3^, mice were randomized as control (vehicle) and EC359 (5 mg/kg/3 days/week/ip for ECa-15-PDX and 10 mg/kg/3 days/week/oral for ECa-81-PDX) treatment groups. The results showed that EC359 treatment significantly reduced the Type II ECa tumor progression compared with the control vehicle-treated group (Figure 5A,B,D,E). The body weights of mice in the vehicle and EC359-treated groups remained unchanged (Figure 5C,F), confirming the low toxicity of EC359. Moreover, EC359-treated tumors exhibited fewer proliferating cells (Ki-67–positive cells) compared with vehicle-treated tumors (Figure 5G,H). These results suggest that EC359 has therapeutic activity in primary PDX tumors.

## 3. Discussion

LIF and LIFR are widely expressed in many solid cancers, and their overexpression is frequently correlated with a bad prognosis for patients [17]. Further, LIF/LIFR signaling in cancer cells has been linked to several oncogenic properties, including proliferation and cell survival, playing a role in immune system and chemoresistance [18,19,20]. Through autocrine and paracrine pathways, tumors exhibit enhanced LIF/LIFR-JAK-STAT3 signaling [21,22,23]. Increased LIF and LIFR expression was connected to poor overall survival (OS) in ECa patients [13]. The findings of this study demonstrate that Type II ECa increased LIF expression, and that this expression is correlated with poor survival in Type II ECa.

Advanced-stage Type II ECa have limited treatment options [9] and a clear need exists for the identification of actionable drivers as new targets for Type II ECa. Despite demonstrating modest efficacy in clinical studies, targeted medications, including bevacizumab, enviroximes, and metformin, did not induce a long-lasting remission in individuals with advanced or recurrent ECa [5,7,8]. It is possible that autocrine loops of cytokines such as LIF/LIFR in the tumor microenvironment induce multiple-cell survival and thus contribute to the low efficacy of the approved therapies. LIF is a frequently upregulated gene in cells that are resistant to paclitaxel and carboplatin, and its presence is associated with poor prognosis in ECa patients [24]. LIF signaling is shown to facilitate communication between tumor cells and fibroblasts and mediates stromal fibroblasts’ pro-invasive activation [25]. The results from this study suggest that LIF/LIFR signaling may represent an actionable target for treating Type II ECa.

We recently created EC359, a first-in-class LIFR inhibitor. Using biophysical assays (SPR, MST, modeling, and biotin pulldown) and CRISPR KO cells, the specificity of EC359 to target LIFR was confirmed [13,15]. The functionality of EC359 was validated using triple-negative breast cancer [15,26] and Type I ECa models [13,14]. Other published studies also demonstrated the efficacy of EC359 in suppressing pancreatic cancer [27,28] and renal cancer cells [29]. In this study, utilizing primary and established Type II ECa cells, patient-derived organoids (PDOs) and patient-derived xenograft (PDX) models, we provided evidence that pharmacological inhibition of the LIF/LIFR axis using EC359 potently reduces Type II ECa cell survival and progression in vivo. Further, our results showed that Type II ECa cells are dependent on functional LIF/LIFR signaling as a knockdown of LIFR using two different siRNAs significantly reduced the cell viability and colony formation of Type II ECa cells.

Tumor suppressor genes p53 and PTEN, and several oncogenes including phosphoinositide 3-kinase (PI3K), protein kinase B (Akt), as well as the mammalian target of rapamycin (mTOR), are altered in the pathophysiology of Type-II ECa [30]. LIF signaling is mediated via the LIF receptor (LIFR) complex that comprises LIFR and glycoprotein 130 (gp130) [11]. Multiple signaling pathways, including STAT3, AKT, MAPK, and mTOR, are activated by LIF via LIFR [11,31,32]. The LIF/LIFR axis is implicated in chemoresistance [23,33,34,35]. Our results showed that the constitutive activation of LIF/LIFR signaling, occurring in Type II ECa cells and LIFR inhibitor EC359, was efficient in reducing the activation of LIFR downstream effectors, including STAT3, AKT, and mTOR.

In summary, this study investigated the role of leukemia inhibitory factor (LIF) signaling through LIF receptor (LIFR) in the progression of Type II ECa. Using a knockdown (KD) of LIFR, the study provided genetic evidence that LIFR is involved in promoting Type II ECa progression. Collectively, our results implicate LIFR inhibitor EC359 as a potential targeted therapy for Type II ECa by inhibiting LIF/LIFR oncogenic signaling.

## 4. Materials and Methods

### 4.1. Cell Culture and Reagents

Human endometrial cancer cell line KLE was purchased from the American Type Culture Collection (ATCC, Manassas, VA, USA) and was maintained as per ATCC guidelines and taken from early passages. This cell line belongs to a high-grade endometrial carcinoma subtype. The p-Akt(S473), Akt, p-mTOR(S2448), mTOR, p-S6(S235/236), S6, p-STAT3(Y705), STAT3 and vinculin antibodies were obtained from Cell Signaling Technology (Beverly, MA, USA). The Ki67 antibody was purchased from Abcam (Cambridge, MA, USA). All secondary antibodies were purchased from Sigma (St. Louis, MO, USA). EC359 was developed by Evestra Inc. (San Antonio, TX, USA) and the detailed synthetic protocol was described in the patent W0 2016/154203 A1.6 [15].

### 4.2. Primary Type II Endometrial Cancer Cells

Primary high-grade type II endometrial cancer cells were established from patient-derived endometrial cancer tissues using a University of Texas Health San Antonio (UTHSA) Institutional Review Board-approved protocol (Table 1). These specimens were de-identified; both the PI and research staff did not have access to clinical linkers or codes. All cell lines were maintained in a humidified chamber with 5% CO_2_ at 37 °C. All the methods involving human tissues were conducted in accordance with the Declaration of Helsinki and the standards defined by UTHSA Institutional Review Board. 

### 4.3. Generation of LIFR-Knockdown Cells

ECa cells were transfected using Lipofectamine RNAiMAX Reagent (Thermofisher, Waltham, MA, USA, cat# 13778150) according to the manufacturer’s protocol; briefly, cells were transfected with either with scrambled-siRNA (siRNA Universal Negative Control #1), LIFR-siRNAs [(siLIFR-1/siLIFR-2; Sigma-Aldrich, St. Louis, MO, USA) (LIFR siRNA ID: SASI_Hs02_00330114 (siLIFR-1), or SASI_Hs02_00330115 (siLIFR-2)] for 60–72 h before being subjected to Western blot analysis, as well as cell viability and colony formation assays.

### 4.4. Cell Viability Assays

The effect of EC359 treatment on the cell viability of Type II ECa cells was assessed using MTT assays as previously described [15]. After seeding Type II endometrial cancer cells in 96-well plates (2 × 10^3^ cells/well), cells were treated with varied concentrations of EC359 for 5 days and cell viability was assessed by adding MTT reagent to the cells. Cells that are actively respiring transform the water-soluble MTT into insoluble purple formazan. The formazan is then dissolved in DMSO, and its optical density is measured using a UV-visible spectrophotometer.

### 4.5. Clonogenic Survival Assays

For clonogenic assays, Type II ECa model cells were seeded at a density of 1000 cells per well in triplicate within 6-well plates. These cells were then treated with vehicle or EC359 for a duration of 5 days, followed by a growth period of 14 days. The cells were immobilized in methanol at a low temperature and subsequently treated with a solution containing crystal violet at a concentration of 0.5% and then counted using ImageJ software Version 1.51 [15].

### 4.6. Cell Invasion Assays

The effect of EC359 on cell invasion of ECa cells was determined using the Corning Bio Coat Growth Factor Reduced Matrigel Invasion Chamber assay (Corning, Corning, NY, USA) [15]. Patient-derived ECa cells were treated with vehicle or EC359 (25 nM) for 22 h and the invaded cells in the treatment conditions were assessed according to the manufacturer’s protocols.

### 4.7. Apoptosis Assays

The effect of EC359 on the apoptosis of Type II endometrial cancer cells was analyzed using the Annexin V/PI kit as per the manufacturer’s instructions (BioLegend, San Diego, CA, USA). Cells were treated with either vehicle or EC359 for 24 h and harvested in Annexin V binding buffer. Then, 100 µL of cell suspension were incubated with Annexin V-FITC and PI for 15 min at room temperature in the dark. Annexin V binding buffer (400 µL) was then added to each sample and stained cells were analyzed using flow cytometry.

### 4.8. Reporter Gene Assays

For STAT3-luc assays, Type II ECa cells were stably transduced with STAT3-firefly luciferase reporter lentiviral particles purchased from Cellomic Technology (Helethrone, MD, USA). STAT3-luc reporter expressing cells was treated with EC359 for 18 h. Cells were lysed in luciferase lysis buffer, and the luciferase activity was measured by the luciferase reporter assay system (Promega, Madison, WI, USA) using a luminometer [15].

### 4.9. Western Blotting and RT-qPCR

Western blotting and RT-qPCR were performed as described in our earlier publication [13]. For Western blotting, cells were lysed in RIPA buffer (Thermo Fisher Scientific, Waltham, MA, USA) containing protease and phosphatase inhibitors. Blots were developed using the ECL kit (Thermo Fisher Scientific, Waltham, MA, USA) and the ChemiDoc Imaging System (BIO-RAD, Hercules, CA, USA). Densitometric analysis was performed using ImageJ software Version 1.51 [15]. Reverse transcription (RT) reactions were performed by using SuperScript III First Strand kit (Invitrogen, Carlsbad, CA, USA), according to manufacturer’s protocol. Real-time PCR was conducted using PowerUp SYBR Green master mix (Applied Biosystems, Foster City, CA, USA) on a CFX96 Real-Time PCR system.

### 4.10. Patient-Derived Organoid (PDO) Studies

Patient-derived organoids (PDOs), generated from de-identified Type II ECa tumor tissues (Obstetrics and Gynecology tissue core), were cultured as previously described [13,14]. For cell viability assays, 5 × 10^3^ cells/10 μL drop were seeded per well of a 96-well plate. Dose range of EC359 or vehicle (DMSO) control was applied to the organoid cultures (in triplicate). Cell viability was assayed after 7 days of treatment using the Promega^®^ CellTiter-Glo^®^ 3D-Superior Cell Viability Assay reagent following the manufacturer’s instructions (Promega, Madison, WI, USA). The intensity of luminescence was measured using a GloMax^®^ Discover System (Promega, Madison, WI, USA) [14].

### 4.11. In Vivo Patient-Derived Xenograft (PDX) Models and Immunohistochemistry (IHC)

All animal experiments were performed using approved UT Health San Antonio IACUC protocol and guidelines. SCID mice were implanted with PDX tumor tissue and tumors were established as previously described [13,14]. Mice were randomized to control or treatment with EC359. Tumor growth was measured using calipers at 3–4-day intervals and volume was calculated using a modified ellipsoidal formula: tumor volume e = ½ (L × W^2^), where L is the longitudinal diameter and W is the transverse diameter. At the end of the experiment, mice were euthanized, and tumors were excised and weighed, fixed in 10% buffered formalin for IHC studies. Tumor sections were incubated with Ki67 primary antibody overnight at 4 °C, followed by secondary antibody incubation for 45 min at room temperature. Immunoreactivity was visualized using the DAB substrate and counterstained with hematoxylin (Vector Lab, Burlingame, CA, USA).

### 4.12. Statistical Analysis

All statistical analyses were performed using GraphPad Prism 9 (GraphPad Software, San Diego, CA, USA). Either a student *t*-test or a one-way or two-way ANOVA were used to assess statistical differences between the control and EC359-treated groups. All the data represented in bar graphs are shown as mean ± SE. A *p*-value of less than 0.05 was considered significant.

## 5. Conclusions

In conclusion, our data indicate that overexpression of LIF signaling is associated with poor survival of Type II ECa and that LIF/LIFR signaling is essential for the survival and progression of Type II ECa. Further, LIFR inhibitor EC359 inhibits LIF/LIFR signaling in Type II ECa cells, lowers cell viability, promotes apoptosis, and reduces in vivo progression. Our findings implicate LIFR inhibitor EC359 as a potential targeted therapy for treating Type II ECa.

## Figures and Tables

**Figure 1 ijms-24-17426-f001:**
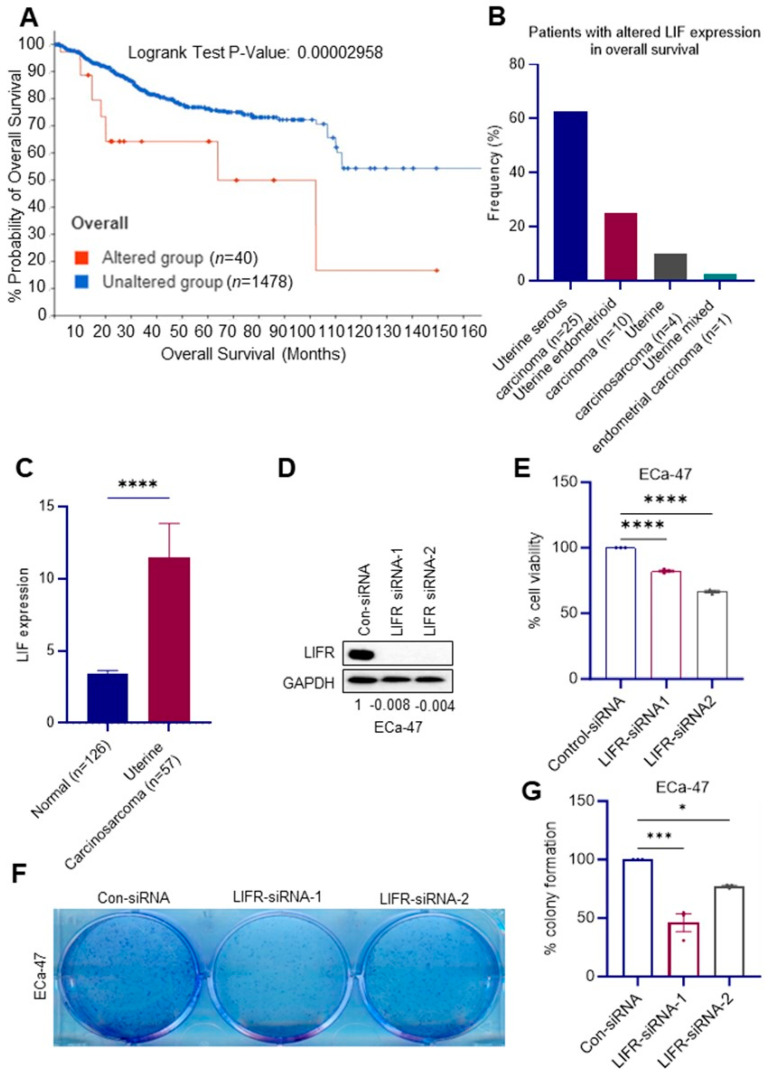
LIF is overexpressed in Type II ECa and Knockdown (KD) of LIFR reduced ECa progression in vitro. (**A**) Association of LIF expression with overall survival of ECa patients (cBioportal). (**B**) Patients with altered LIF expression in overall survival patient cohort. (**C**) Expression of LIF in normal (*n* = 126) and uterine carcinosarcoma (*n* = 57) from TCGA patient cohort (OncoDB). (**D**) Levels of LIFR in siRNA-mediated KD patient-derived ECa cells were measured via Western blotting. Densitometric analysis of Western blots was performed using ImageJ software Version 1.51 and the fold change was calculated using total LIFR protein over GAPDH protein expression. (**E**) The effect of LIFR-KD on cell viability in ECa-47 cells was measured using CellTiter-Glo assays. (**F**) The effect of LIFR-KD on ECa-47 cell survival was measured using colony formation assays. Quantitation is shown in the right panel (**G**). Data in (**D**–**G**) are representative of three independent experiments (*n* = 3). The *p*-values in (**C**,**E**,**G**) were calculated using one-way ANOVA. Data are represented as mean ± SE. * *p* < 0.05; *** *p* < 0.001; **** *p* < 0.0001.

**Figure 2 ijms-24-17426-f002:**
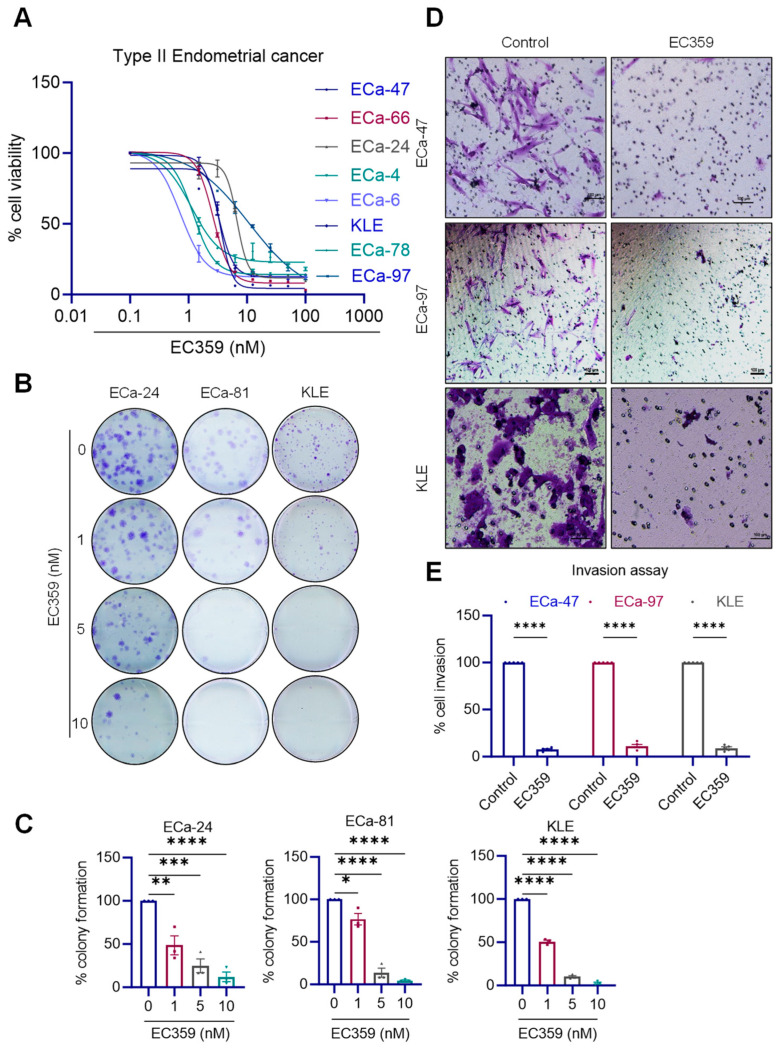
LIFR inhibitor EC359 reduced cell viability, colony formation and invasion of Type II ECa cells. (**A**) Effect of EC359 on cell viability of established and patient-derived primary ECa cells was determined using MTT assay. (**B**) Effect of EC359 on cell survival of ECa cells was measured using colony formation assay. (**C**) Quantitation of the percentage of colonies is shown. The effect of EC359 on cell invasion of Type II ECa cells was determined using Boyden chamber assays. Images were shown in panel (**D**) and quantitation of the percentage of cells invaded is shown in panel (**E**). Data are representative of three independent experiments (*n* = 3). Data are represented as mean ± SE. Scale bar represents 100 µm. In (**C**,**E**), *p*-values were calculated using one-way ANOVA. * *p* < 0.05; ** *p* < 0.01; *** *p* < 0.001; **** *p* < 0.0001.

**Figure 3 ijms-24-17426-f003:**
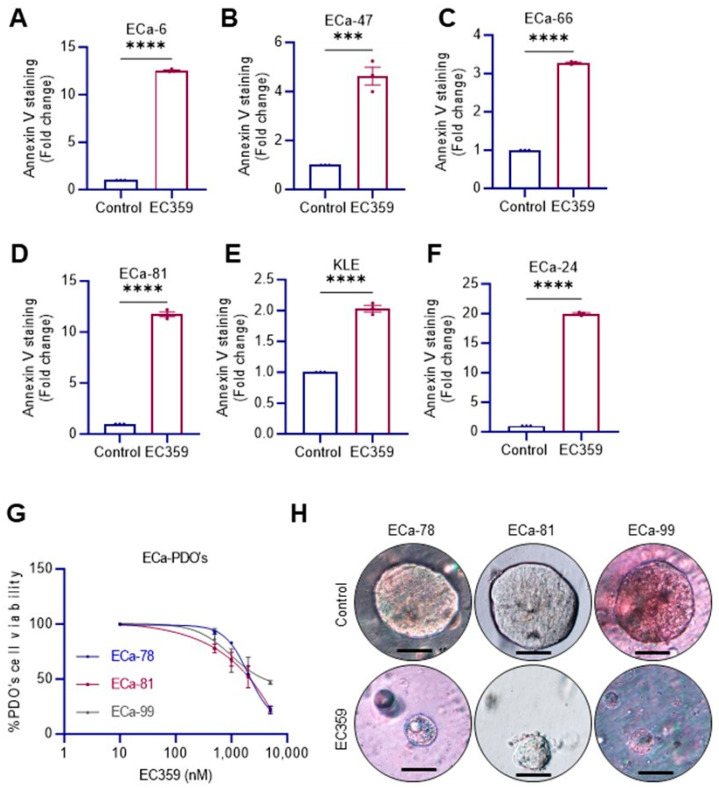
LIFR inhibitor EC359 promoted apoptosis of Type II ECa cells and reduced the growth of PDOs. (**A**–**F**) Effect of EC359 (100 nmol/L) on apoptosis of patient-derived type II ECa cells (*n* = 3) was determined using Annexin V staining. (**G**) Effect of various doses of EC359 treatment in patient-derived organoids (PDOs) was measured using CellTiter-Glo luminescent cell viability assay. (**H**) PDO images were shown. Scale bar represents 100 µm. Data are representative of three independent experiments (*n* = 3). Data are represented as mean ± SE. In (**A**–**F**), *p*-values were calculated using student’s *t*-test. *** *p* < 0.001; **** *p* < 0.0001.

**Figure 4 ijms-24-17426-f004:**
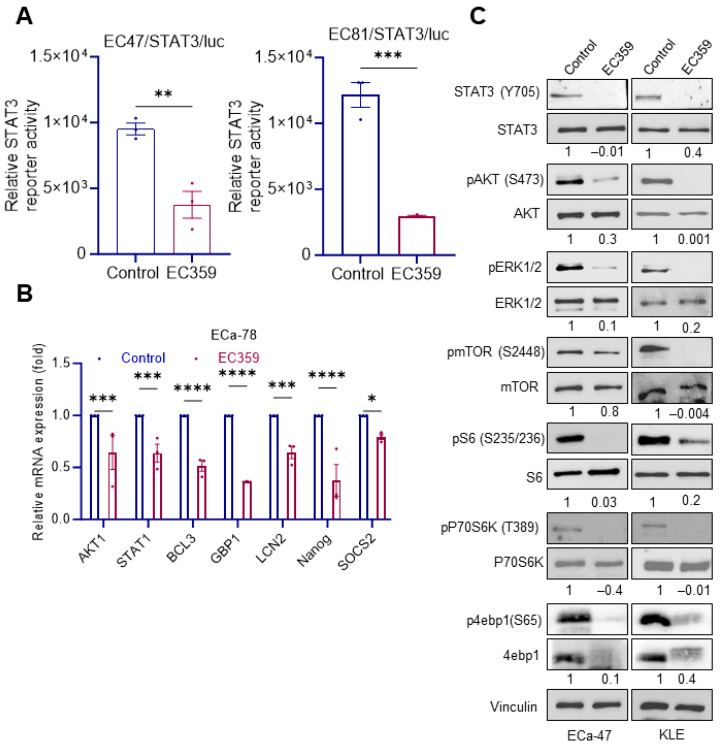
EC359 inhibits LIFR downstream signaling in Type II ECa cells. (**A**) ECa-47, and ECa-81 cells stably expressing STAT3-luc reporter were treated with EC359 (50 nmol/L) and reporter activity was measured after 18 h. (**B**) Effect of EC359 (100 nmol/L) treatment (6 h) on STAT3 target genes was measured using RT-qPCR analysis (*n* = 3) in ECa-78 cells. (**C**) ECa-47 and KLE cells were treated with EC359 (100 nmol/L) for 6 h and the status of LIFR downstream signaling was measured using Western blotting. Densitometric analysis of Western blots were performed using ImageJ software Version 1.51 and the fold change was calculated using the phosphorylated protein over total protein expression of each protein. Data are representative of three independent experiments (*n* = 3). Data are represented as mean ± SE. In (**A**,**B**), *p*-values were calculated using student’s *t*-test and one-way ANOVA, respectively. * *p* < 0.05; ** *p* < 0.01; *** *p* < 0.001; **** *p* < 0.0001.

**Figure 5 ijms-24-17426-f005:**
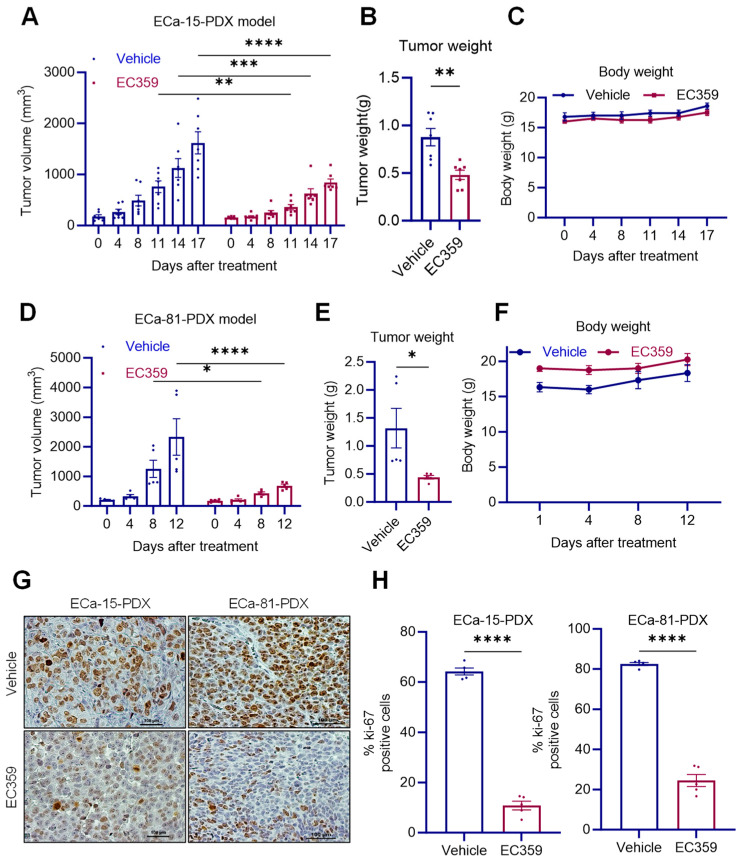
EC359 inhibits the growth of Type II ECa xenograft tumors. Patient-derived xenograft (PDX) tumors were treated with vehicle or EC359 (5 mg/kg/3 days/week/ip for ECa-15-PDX and 10 mg/kg/3 days/week/oral for ECa-81-PDX). EC359 was administered intraperitoneally (i.p.) to ECa-15 xenograft tumor-bearing mice and orally (gavage) to ECa-81 xenograft tumor-bearing mice. Tumor volumes and tumor weights and body weights of ECa-15 (*n* = 7 tumors) (**A**–**C**) and ECa-81 (*n* = 5 tumors) (**D**–**F**) are shown. (**G**) Ki-67 expression as a marker of proliferation was analyzed via IHC and quantitated (**H**). Data are represented as mean ± SE. *p*-values were calculated using student’s *t*-test (**B**,**E**,**H**) and two-way ANOVA (**A**,**D**). * *p* < 0.05; ** *p* < 0.01; *** *p* < 0.001; **** *p* < 0.0001.

**Table 1 ijms-24-17426-t001:** List of primary patient-derived Type II ECa samples used for the study.

Number	Histology	Grade
ECa-4	Serous Papillary	Grade 3
ECa-6	Serous Papillary	Grade 3
ECa-15	Carcinosarcoma-assoc serous papillary carcinoma	Grade 3
ECa-24	Endometrioid & Serous	Grade 3
ECa-47	Serous papillary carcinoma	Grade 3
ECa-66	Serous papillary endometrial cancer	Grade 3
ECa-78	Clear Cell	Grade 3
ECa-81	Carcinosarcoma	Grade 3
ECa-97	Carcinosarcoma	Grade 3
ECa-99	Carcinosarcoma	Grade 3

## Data Availability

The listed web links http://cbioportal.org/ (accessed on 8 October 2023); http://www.oncodb.org/ (accessed on 8 October 2023) were used to collect the data for Figure 1A–C.

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
