# Peer review of "The LIFR Inhibitor EC359 Effectively Targets Type II Endometrial Cancer by Blocking LIF/LIFR Oncogenic Signaling"

_ijms, 2023, doi:10.3390/ijms242417426_

Round 1

Reviewer 1 Report

Comments and Suggestions for Authors

The manuscript "Therapeutic Utility of EC359 for Targeting Oncogenic LIF/LIFR 2 Signaling in Type II Endometrial Cancer" presents intriguing research into a novel therapeutic approach for a particularly aggressive cancer subtype. However, the study is undermined by several significant shortcomings that span across its various sections. These issues range from methodological limitations to a lack of comprehensive data analysis and insufficient contextual and mechanistic insights. Addressing these shortcomings is crucial to enhance the credibility and scientific value of the research.

Comments:

1. The manuscript titled "Therapeutic Utility of EC359 for Targeting Oncogenic LIF/LIFR 2 Signaling in Type II Endometrial Cancer" presents an intriguing exploration of a novel therapeutic approach for an aggressive cancer subtype. However, it lacks specificity in its title regarding EC359's molecular action. A title that elaborates on EC359's mechanism or its specific target within the LIF/LIFR pathway would provide more clarity and direct the reader towards the study's primary focus.

2. The abstract, while comprehensive in scope, falls short in providing specific data or statistical outcomes to substantiate the claims made. This omission detracts from the abstract's potential to effectively summarize and highlight the study's significance and its findings. Including key data or results would lend greater weight to the abstract and better orient the reader to the study's conclusions.

3. Transitions between topics within the abstract are abrupt and disjointed. This lack of smooth flow disrupts the reader’s understanding of the research's scope and aims. Smoother transitions between the problem statement, research approach, and findings would significantly enhance the clarity and comprehensibility of the abstract.

4. In the introduction, the context provided for the significance of researching Type II endometrial cancer (ECa), especially in comparison to Type I, is insufficient. This lack of depth fails to highlight the urgency and need for this specific research. A more detailed contextual background would help readers understand why this study is crucial and timely. Furthermore, it's surprising that you neglected to incorporate some of the recent pivotal research related to cancer genomic analysis [1-3].

5. The novel aspects of this study, particularly how EC359 differs from or improves upon existing treatments for Type II ECa, are not emphasized enough in the introduction. Highlighting the unique features and potential advantages of EC359 is essential to underscore the study's relevance and innovation in the field.

6. The introduction includes references to previous studies but lacks a critical examination of these works. A discussion on how this study builds upon or challenges existing literature would provide a more robust background and position the current study within the broader research landscape more effectively.

7. The introduction would benefit from a brief overview of the methodologies employed in the study. Providing readers with a clearer expectation of the study's approach, design, and methods would aid in understanding the structure and direction of the research.

8. The introduction mentions the unexplored role of LIF/LIFR in Type II ECa but stops short of hypothesizing or providing a rationale for its potential significance. This gap in the foundational rationale leaves readers with unanswered questions about the study's underlying hypotheses and theoretical framework.

9. The introduction's connection between the study's findings and their potential clinical implications in the treatment of Type II ECa is not made explicit. Detailing this connection is vital for illustrating the real-world impact and clinical relevance of the research.

10. Both the abstract and introduction lack specific data or anticipated outcomes of the study. Including more detailed information about the study's scope, data, and potential impact would provide a clearer and more comprehensive picture of the research's significance and expected contributions to the field.

11. The presentation of data in the results section, particularly in the figures, is not adequately detailed in the accompanying text. This lack of clarity makes it challenging for readers to fully understand and interpret the study's findings. Clear and detailed descriptions of the figures are essential for the reader to grasp the implications of the results.

12. The results section includes basic statistical significance indicators but does not delve into a comprehensive statistical analysis. This superficial treatment of statistics undermines the credibility and robustness of the findings. A more thorough statistical analysis would strengthen the validity and reliability of the results presented.

13. Key procedures in the study, such as Western blotting and cell viability assays, are described without sufficient detail, impacting the study's reproducibility. Providing specific procedural details is crucial for enabling other researchers to replicate and validate the findings, which is essential for the study's acceptance and credibility in the scientific community.

References:

[1] Huang, Qinghua, Dan Wang, Zhenkun Lu, Shichong Zhou, Jiawei Li, Longzhong Liu, and Cai Chang. "A novel image-to-knowledge inference approach for automatically diagnosing tumors." Expert Systems with Applications 229 (2023): 120450.

[2] Wang, Dan, Yingchen Wang, Yongzhen Wang, Longzhong Liu, Jiawei Li, and Qinghua Huang. "Fusion of Human Cognitive Knowledge and Machine Inference for Breast Cancer Detection." In 2023 International Conference on Advanced Robotics and Mechatronics (ICARM), pp. 179-184. IEEE, 2023.

[3] Xi, Jianing, Dan Wang, Xuebing Yang, Wensheng Zhang, and Qinghua Huang. "Cancer omic data based explainable AI drug recommendation inference: A traceability perspective for explainability." Biomedical Signal Processing and Control 79 (2023): 104144.

Comments on the Quality of English Language

Moderate editing of English language required.

Reviewer 2 Report

Comments and Suggestions for Authors

In the manuscript titled “Therapeutic Utility of EC359 for Targeting Oncogenic LIF Signaling in Type II Endometrial Cancer, Authors proposed LIFR inhibitor EC359 as a potential novel small molecule therapeutics for the treatment of Type II ECa. Authors performrd multiple biochemical assays and validated role of EC359 to attenuate the activation of LIF/LIFR driven pathways including STAT3, mTOR, and AKT in patient derived xenograft models. The manuscript is very interesting. However, there are few major revisions as following:

1. The authors need to provide scales bares for all the photographic/IHC figures.

2. Please provide number of replicates in fig 1d, 4c,

3. Please provide fold change in 1d, 4c by using densitometric analysis.

4. Please provide number of replicates in figures 1, 2b, 2c, 3d,3e, 4d,4e.

5. GAPDH blot in 1d is too saturated. Please provide lighter exposure for the image.

6. The classical activation of apoptosis requires caspase/parp activation. Authors need to check levels of caspase/parp-1 cleavage in experimental conditions in Fig 3.

7. Screening for mTOR activation markers is not sufficient in figure 4C. Authors need to check levels of p-4EBP1(Ser65)/ 4-EBP1 , p-p70S6K(Thr389)/ p70S6K in experimental conditions in figure 4C.

Minor comment:

8. The manuscript requires a significant attention to improve punctuations, grammar and the readability.

Comments on the Quality of English Language

please see comment 8 above.

Round 2

Reviewer 1 Report

Comments and Suggestions for Authors

I have no more question.

Comments on the Quality of English Language

Moderate editing of English language required.

Reviewer 2 Report

Comments and Suggestions for Authors

In the updated manuscript Therapeutic Utility of EC359 for Targeting Oncogenic LIF Signaling in Type II Endometrial Cancer”, the authors did address all the previous concerns and now the manuscript is convincing and would help advance the understanding of role of LIFR inhibitor EC359 as a potential novel small molecule therapeutics for the treatment of Type II ECa. The manuscript is now updated, and I recommend this article for publication.